

# Relying on known or exploring for new? Movement patterns and reproductive resource use in a tadpole-transporting frog

Kristina B. Beck[1,2], Matthias-Claudio Loretto[2], Max Ringler[3,4], Walter Hödl[4] and Andrius Pašukonis[2,5]

[1] Department of Behavioural Ecology and Evolutionary Genetics, Max Planck Institute for Ornithology, Seewiesen, Germany
[2] Department of Cognitive Biology, University of Vienna, Vienna, Austria
[3] Department of Ecology and Evolutionary Biology, University of California, Los Angeles, CA, United States of America
[4] Department of Integrative Zoology, University of Vienna, Vienna, Austria
[5] FAS Center for Systems Biology, Harvard University, Cambridge, MA, United States of America

Corresponding author
Kristina B. Beck, kbeck@orn.mpg.de

## ABSTRACT

Animals relying on uncertain, ephemeral and patchy resources have to regularly update their information about profitable sites. For many tropical amphibians, widespread, scattered breeding pools constitute such fluctuating resources. Among tropical amphibians, poison frogs (Dendrobatidae) exhibit some of the most complex spatial and parental behaviors—including territoriality and tadpole transport from terrestrial clutches to ephemeral aquatic deposition sites. Recent studies have revealed that poison frogs rely on spatial memory to successfully navigate through their environment. This raises the question of when and how these frogs gain information about the area and suitable reproductive resources. To investigate the spatial patterns of pool use and to reveal potential explorative behavior, we used telemetry to follow males of the territorial dendrobatid frog *Allobates femoralis* during tadpole transport and subsequent homing. To elicit exploration, we reduced resource availability experimentally by simulating desiccated deposition sites. We found that tadpole transport is strongly directed towards known deposition sites and that frogs take similar direct paths when returning to their home territory. Frogs move faster during tadpole transport than when homing after the deposition, which probably reflects different risks and costs during these two movement phases. We found no evidence for exploration, neither during transport nor homing, and independent of the availability of deposition sites. We suggest that prospecting during tadpole transport is too risky for the transported offspring as well as for the transporting male. Relying on spatial memory of multiple previously discovered pools appears to be the predominant and successful strategy for the exploitation of reproductive resources in *A. femoralis*. Our study provides for the first time a detailed description of poison frog movement patterns during tadpole transport and corroborates recent findings on the significance of spatial memory in poison frogs. When these frogs explore and discover new reproductive resources remains unknown.

## INTRODUCTION

In a dynamic environment resource availability changes in time and space which has major influences on animal movement decisions (*Milner-Gulland, Fryxell & Sinclair, 2011*; *Bell, 2012*). Animals that rely on unpredictable, ephemeral, and patchy resources have to explore their environment regularly (*Roshier, Doerr & Doerr, 2008*). Updating information on profitable resources can be achieved by exploring unknown areas to collect new information, by frequently visiting already known patches to affirm the availability of resources, or by prospecting for new resources within familiar areas (*Real, 1981*; *Eliassen et al., 2009*; *Díaz et al., 2013*). However, exploration comes at a cost of the time spent searching, which conflicts with other fitness-related activities such as territory defense (cf. *Ydenberg & Krebs, 1987*) and advertising for mates (*Thomas et al., 2003*). Exploration also increases the exposure to sit-and-wait predators and thus overall predation risk (*Stamps, 1995*). Thus, animals depending on fluctuating resources need to find a balance between relying on known resources and prospecting for new ones (*Milner-Gulland, Fryxell & Sinclair, 2011*).

Most amphibians depend on aquatic sites for breeding, making this taxon particularly suitable for investigating how animals deal with varying availability, stability and distribution of resources. Amphibians exhibit a great diversity of reproductive strategies ranging from explosive breeders that gather at large ponds for synchronized spawning, to prolonged breeders with terrestrial clutches that use small widespread pools for tadpole development (*Duellman & Trueb, 1994*; *Crump, 2015*). Tropical amphibians, in particular, are renowned for the variety of aquatic sites that they use for reproduction (*Wells, 2007*). Despite the overall large amount of rainfall, seasonal variability can have a strong impact on the availability of breeding resources and thus on the reproductive success of tropical amphibians (*Aichinger, 1987*; *Bertoluci & Rodrigues, 2002*; *Gottsberger & Gruber, 2004*). During heavy rainfall, potential breeding sites such as small pools in the ground can appear within hours. However, desiccation also happens rapidly due to the high environmental temperatures. To date, it remains mostly unknown how tropical amphibians deal with the uncertainty of breeding sites, and which mechanisms they use to find these scattered resources.

The Neotropical poison frogs (Dendrobatidae) exhibit a variety of complex spatial behaviors such as territoriality, tadpole transport, and offspring provisioning (*Weygoldt, 1987*; *Pröhl, 2005*; *Lötters et al., 2007*). The complex parental behavior in this group of frogs has attracted a considerable amount of research (e.g., *Brown, Morales & Summers, 2008*; *Dugas et al., 2015*; *Ringler et al., 2015*; *Schulte & Summers, 2017*; for reviews, see *Wells, 2007*; *Summers & Tumulty, 2013*; *Roland & O'Connell, 2015*). However, the associated movement patterns and the factors shaping them have rarely been quantified and remain poorly understood (but see *Summers, 1990*; *Brown, Morales & Summers, 2009*;

*Ringler et al., 2013*). Most poison frogs show site fidelity to their territory and shuttle their larvae from terrestrial clutches to widespread, ephemeral deposition sites, where the larvae complete their development (e.g., *Ameerega trivittata* and *A. femoralis* (*Roithmair, 1992*); *Ranitomeya imitator* and *R. variabilis* (*Brown, Morales & Summers, 2009*); *R. reticulata* (*Werner et al., 2010*); *Colostethus panamansis* (*Wells, 1980*); *Dendrobates auratus* (*Summers, 1990*); *D. leucomelas* and *Oophaga histrionica* (*Summers, 1992*); for reviews, see *Weygoldt, 1987*; *Pröhl, 2005*; *Wells, 2007*). When transporting their offspring, frogs need to know when and where to go for suitable water bodies that are persistent enough to allow larval development, but ephemeral enough to minimize predator abundance (cf. *Murphy, 2003*; *Lehtinen, 2004*). Recent studies have shown that poison frogs can use flexible learning strategies in spatial tasks in captivity (*Dendrobates auratus* (*Liu et al., 2016*)) and rely on prior experience to successfully return home after translocation (*Allobates femoralis* (*Pašukonis et al., 2013*; *Pašukonis et al., 2014a*)) and to find tadpole deposition sites in the field (*Pašukonis et al., 2016*). Further, two poison frog species have been shown to use spatial rather than direct cues for offspring recognition (*Stynoski, 2009*; *Ringler et al., 2016b*). Together these results suggest that poison frogs rely on spatial memory to successfully navigate in their environment, which raises the question of when and how these frogs gain information about the area and in particular how they update information on the availability of their reproductive resources. To date, exploratory behavior has only been reported in two poison frog species in the form of prospecting trips after tadpole deposition, between tadpole transport events (*D. auratus* (*Summers, 1989*)) or prior to tadpole deposition (female *Oophaga pumilio* (*Brust, 1990*)). However, the detailed movements during transport and exploration have not been quantified for any poison frog species. In this study, we attempted to fill this knowledge gap by tracking the well-studied poison frog *Allobates femoralis* during tadpole transport.

*Allobates femoralis* is a small territorial poison frog with predominantly male tadpole transport (*Ringler et al., 2013*). Males spread their tadpoles across multiple, scattered aquatic sites; it has been suggested that this approach increases offspring survival (*Erich et al., 2015*). Their dependency on unpredictable resources and experience-based navigation make *A. femoralis* an ideal study species to address questions concerning the mechanisms and timing of environmental exploration and learning. The demonstrated ability of *A. femoralis* to navigate to locations with high spatial precision (*Pašukonis et al., 2014a*; *Pašukonis et al., 2014b*; *Pašukonis et al., 2016*) makes it easier to interpret their movement patterns, as it suggests that non-directed movement in the local area is more likely to be exploratory behavior than an inability to orient in space. Since tadpole transport constitutes the most prominent long-distance movement in *A. femoralis* (*Ringler, Ursprung & Hödl, 2009*; *Ringler et al., 2013*) it has been postulated that the frogs update their knowledge about the area by exploring during tadpole transport and subsequent homing (*Pašukonis et al., 2013*; *Pašukonis et al., 2014b*).

During the tadpole transport male frogs usually leave their territory, which provides them with a chance to gain information on resource location and quality. However, such exploration during tadpole transport would result in a trade-off between the potential benefits gained and the costs related to searching behavior. During the breeding season,

male *A. femoralis* are mostly found in their territories, calling to attract females and to repel competing males (*Kaefer et al., 2012*). Leaving the territory to explore would increase the risk of losing mating opportunities, losing the entire territory, and can increase energetic expenditure as well as the risk of predation (e.g., *Wolf, Hainsworth & Gill, 1975*; *Townsend, 1986*; *Roithmair, 1992*). Tadpole transport consists of two phases: first, shuttling the tadpoles until deposition, then homing back to the territory. Exploratory behavior on the way to known deposition sites would also incur potential costs for the offspring being transported, such as increased risk of desiccation (*Downie et al., 2005*), whereas during homing it would only incur costs for the male. Thus, the net benefit of prospecting for new deposition sites during homing should be higher, which might be reflected in frog movement patterns.

In this study we quantified the movement patterns associated with tadpole transport and factors that potentially influence them: attraction to cues originating from the pools, pool desiccation, and the weather. We used telemetry to follow transporting male *A. femoralis* towards artificial deposition sites and back to their territory. To examine whether the frogs performed any exploratory behavior we attempted to induce exploration by removing artificial pools to simulate desiccation. We predicted that tadpole carriers would show fast, directional movement to known pools, as we expected them to aim at reducing potential costs for the transported offspring and to tend to perform exploratory detours on the way back to their territories. Further, we expected frogs encountering a location where a pool had been removed before deposition to be more likely to perform exploratory detours and to continue visiting other deposition sites to update their information on pool availability.

## MATERIALS & METHODS

### Study species and area

*Allobates femoralis* is a small diurnal frog (snout-urostyle length approximately 25 mm) common throughout Amazonia and the Guiana Shield (*Amézquita et al., 2009*). During the rainy season males occupy territories (average defended area: 151.13 m$^2$ (*Ringler et al., 2011*)) which they advertise by calling and defend for up to several months (*Roithmair, 1992*; *Ringler, Ursprung & Hödl, 2009*). Mating and oviposition of approximately 20 eggs take place in the leaf litter inside the male's territory (*Roithmair, 1992*; *Ringler et al., 2012*). After 15–20 days of development, the male revisits the clutch, allows the tadpoles to wriggle onto his back, and transports them to widely distributed deposition sites. On average, males are found with eight tadpoles when transporting (range 1–25, cf., *Ringler et al., 2013*), and they can deposit them at several pools (*Erich et al., 2015*). Females will transport the offspring only when males disappear (*Ringler et al., 2015*). The frogs use a variety of small to medium-sized terrestrial water bodies such as rain-flooded depressions, holes in fallen trees, or palm bracts for tadpole deposition, and recent tracking revealed that *A. femoralis* remember the location of at least six different pool sites (*Pašukonis et al., 2016*). Aquatic deposition sites constitute a limiting resource for *A. femoralis,* and frogs readily use artificial pools when provided in their natural habitat (*Ringler, Hödl & Ringler, 2015*).

We carried out the study from 18 January–12 March 2015 in a lowland rainforest on a five hectare river island near the "Camp Pararé" field site at the CNRS "Nouragues

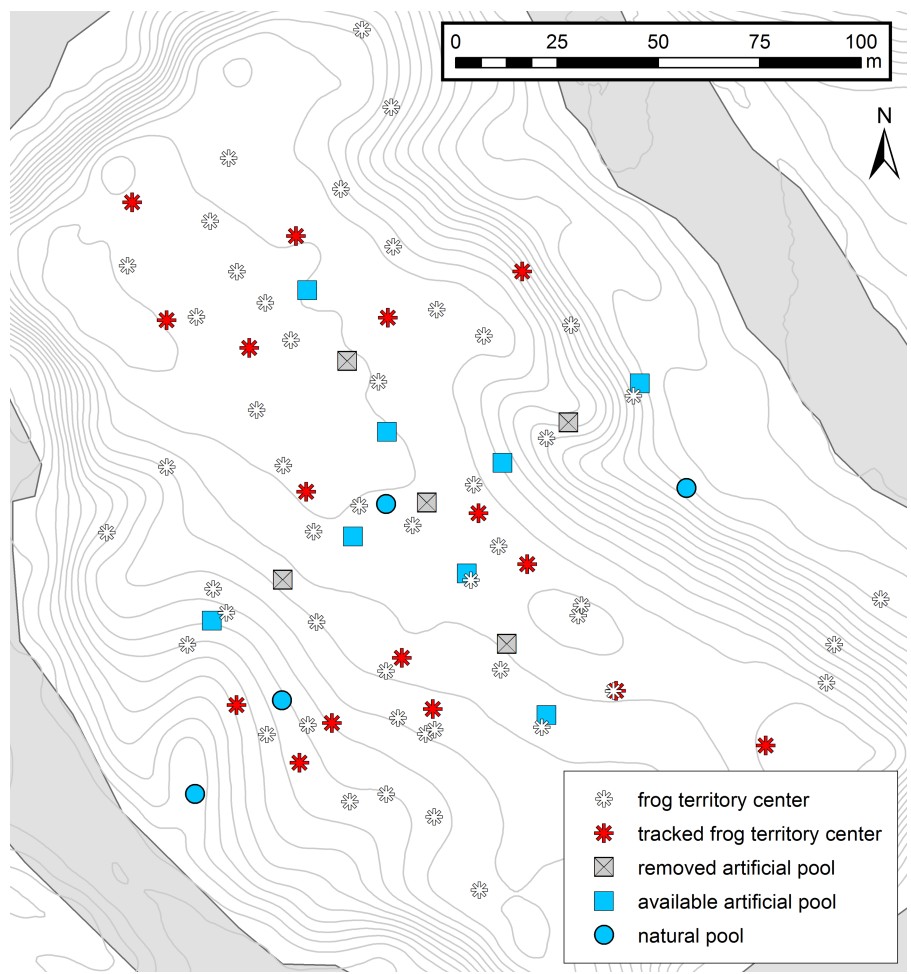

**Figure 1** **Map of the study area.** Experimental setup and the distribution of male territories in the study area. Red asterisks represent the center of tracked frog territories ($n = 16$) and white asterisks represent the territory centers of other identified males in the area ($n = 49$). Squares represent the cross-array of thirteen artificial tadpole deposition sites, blue squares representing available pools and gray crossed squares the removed deposition sites. Blue circles represent four potential natural pools, which were visited by tadpole carriers during tracking. Contour lines (1 m) and the Arataye River are drawn in light gray.

Ecological Research Station'' in the Nature Reserve ''Les Nouragues'', French Guiana (3°59′N, 52°35′W) (*Bongers et al., 2013*; *Ringler et al., 2016a*). All necessary permits were provided by the Centre National de la Recherche Scientifique (CNRS) and by the Direction Régionale de l'Environnement, de l'Aménagement et du Logement (DEAL: ARRETE no 2011-44/DEAL/SMNBSP/BSP). The island population was established by introducing 1800 genotyped *A. femoralis* tadpoles from a nearby population in 2012 (*Ringler, Mangione & Ringler, 2014*) that were released in artificial pools (volume ~12 l, inter-pool distance ~10 m). In 2013, the pools were rearranged in a cross-shaped array of 13 pools (inter-pool distance ~20 m). At the beginning of our study in 2015 we removed every second pool to experimentally simulate dried-up water bodies, leaving 8 pools available (Fig. 1). Occasionally, tadpole carriers also used natural deposition sites such as small flooded

depressions and burrows on the forest floor. In total, we recorded four such natural sites which temporarily filled with water, depending on the weather conditions, and which were visited by tadpole-transporting frogs during our study (Fig. 1).

## Territory sampling

To determine male territories, we continuously scanned the area for calling males during our study. All frogs were caught in transparent plastic bags, photographed and individually identified by their unique ventral coloration pattern (*Ringler, Mangione & Ringler, 2014*) using the pattern matching software Wild-ID (*Bolger et al., 2011*). We determined sex based on calling behavior and the presence (male) or absence (female) of a vocal sac. We recorded exact capture positions on a detailed GIS background map (*Ringler et al., 2016a*) using tablet PCs (WinTab 8, Odys) with a mobile GIS software (ArcPad 10, ESRI). To calculate the center of the territory for each male, we only used data points where males displayed territorial behavior (calling, courtship, aggressive approach).

## Tadpole carrier tracking

Because *A. femoralis* clutches are difficult to find and the timing when the male picks up the tadpoles is variable (S Weinlein, pers. obs., 2014–2015), we focused our search effort on finding frogs already transporting tadpoles, mostly by searching around known deposition sites. Sampling was done every day between 07:00 and 13:00 h as tadpole transport mainly occurs in the morning (*Aichinger, 1987*; *Ringler et al., 2013*). We caught the tadpole carriers in transparent plastic bags, photographed them for identification, and recorded their exact position on the GIS map. We counted the number of transported tadpoles, and when some tadpoles fell off during catching and handling, we placed them on the male's back again.

Before the release, we equipped transporting males with a transponder attached to a waistband (Fig. S1). The entire procedure took a few minutes and did not disrupt tadpole transport or deposition behavior regardless of whether the tadpoles were manipulated or not. We followed tagged tadpole carriers using the harmonic direction-finding (HDF) telemetry technique. This system consists of a passive reflector/transponder, which is attached to the animal and an active directional transceiver, which emits and then receives the reflected radio signal. It allows smaller animals to be tracked than would be possible by conventional active radio tracking (*Mascanzoni & Wallin, 1986*; *Rowley & Alford, 2007*), and it has been successfully used in *A. femoralis* (*Pašukonis et al., 2014a*; *Pašukonis et al., 2014b*). We fitted the tags using a silicon tube 2 mm in diameter, forming a waistband with an additional strap between the hind legs to prevent the tag from rotating (Fig. S1). Both parts were fixed with a cotton thread, which would break and release the waistband after approximately two to three weeks (K Beck & A Pašukonis, pers. obs., 2014–2015) in case an individual was not recaptured. The waistbands carried a small diode beneath a color-coded seal and a T-shaped dipole antenna made of flexible, coated wire. The long end (~12 cm) of the antenna dragged freely behind the moving frog while the short end (~2 cm) was secured inside the waistband. As *A. femoralis* is strictly diurnal, we only tracked frogs during daylight hours (07:00 and 19:00 h), relocating each individual and recording their position every 30–60 min. We followed each frog until all tadpoles had

been deposited and the male had returned to his territory, where the tag was removed. We assumed that the frogs had returned when they approached locations at which they had previously displayed territorial behavior. During tracking we tried to minimize disturbance by carefully approaching the signal source while searching with the transceiver until the frog was visually spotted or the origin of the signal could be narrowed down to less than 1 m. We approached individual frogs from different directions so as to not influence or bias their movement in any direction. Occasional disturbances during tracking only influenced our measurements of the directionality of long distance movements minimally, because *A. femoralis* responds to disturbance by immediately hiding in the leaf litter rather than fleeing over longer distances (K Beck, M Ringler & A Pašukonis, pers. obs., 2006–2017). We caught tagged individuals when they had not moved for more than a day to check for possible issues such as skin injuries—in a single case we immediately removed the tag. In one case without movement for more than two days but without apparent injuries we also removed the tag to minimize any potential long-term effects on behavior.

## Tracking data

We handled and visualized spatial data in the GIS software ArcGIS10 (ESRI) after projecting (UTM-zone 22N, WGS1984), and all analyses were performed in the statistical software R version 3.2.0 (*R Development Core Team, 2014*). We split the full trajectories of tracked frogs into tadpole transport (TT) from the first encounter point to the last deposition site, and homing trajectories (HT), from the last deposition site back to the home territory. Movement distance, duration, and speed were calculated for each TT and HT. For individuals that took more than one day to complete deposition and/or homing, we excluded the nights (−12 h per night) to estimate the average time they were moving during tadpole transport. In most cases, the observed TT did not cover the full TT as the frogs were encountered on their way to the pools or in their close vicinity. We excluded individuals encountered immediately before tadpole deposition (frogs already present at a pool or individuals with fewer than two tracking locations recorded before reaching the pool) from the analysis of TT trajectories. We interpolated the total distances of TTs by approximating the missing part from the territory center until the first encounter location by a straight-line to obtain minimum-distance estimates (see Fig. 2).

### Influence of tadpoles on movement speed

We created a generalized linear mixed model (GLMM) to test the effects of tadpole presence (TT vs. HT) on the movement speed of frogs (family = gamma, link = logit). As response variable, we used the average speed per TT and HT, as explanatory variable "tadpoles present" (yes/no), and individual frog ID as random factor to account for repeated trajectories of the same individuals.

### Influence of weather on movement speed

Since frog activity varies throughout the day and depends on weather conditions (*Bellis, 1962*; *Brooke, Alford & Schwarzkopf, 2000*), we investigated potential effects of the weather on the frogs' movement speed using temperature and rainfall measurements, both obtained from an above-canopy weather station (Nouraflux: rainfall sensor Campbell ARG100,

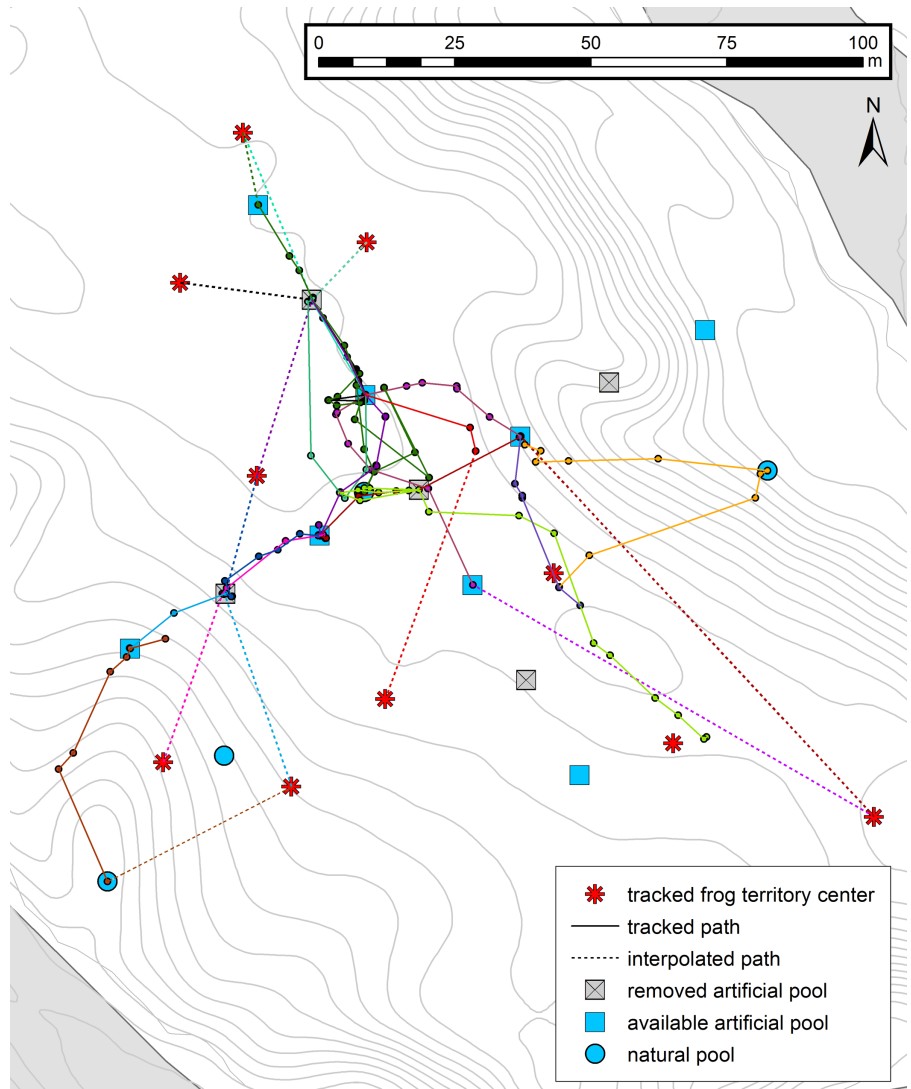

**Figure 2** **Tadpole transport trajectories.** Trajectory map showing movement patterns of tadpole transporting males to and between the deposition sites. Different colors represent different tracking events ($n =$ 15, 10 individuals), full lines represent the frog paths obtained by interpolating the consecutive frog locations and dashed lines the missing path from the territory to the first encounter point. For all other symbols, see Fig. 1.

temperature sensor Vaisala HMP155A). In contrast to the previous model, where the focus was on certain trajectory sections (TT and HT), we focused on 3 h intervals here to investigate variations in the frogs' activity (here: movement speed). The 3 h intervals represent four potentially different daily activity periods in *A. femoralis*, with tadpole transport happening predominately in the morning, low activity around noon, high calling activity during the afternoon, and high calling activity and most female-male interactions (e.g., courtship) in the evening (personal observation by all authors; see also *Kaefer et al., 2012*; *Ringler et al., 2013*). We used a GLMM (family = gamma, link = logit), with speed as response variable, and the explanatory variables "tadpoles present" (yes/no),

"time of the day" (split in four 3 h intervals: 07:00–10:00, 10:00–13:00, 13:00–16:00, 16:00–19:00 h), average "temperature" for each corresponding interval, and cumulative "rain" in millimeters during each interval. Since this analysis is not within the main focus of this study, we present the results in the Tables S1 and S2.

## Pool visits

We defined a pool visit as frogs actually entering a potential deposition site (position either on/in a filled pool or on the removed/dried out pool location). For each tadpole carrier, we recorded all potential tadpole deposition sites visited (available/removed artificial pool or available/dried-out natural site), and whether tadpole deposition occurred at the available pools or not. To test whether frogs explored further pools after the complete deposition of their offspring, we recorded for each location whether there were tadpoles still present on their back or not. If frogs moved to further pools without tadpoles on their back, we considered this to be exploratory behavior. In addition, we calculated the average number of deposition sites the frogs visited (available/removed artificial pool or available/dried-out natural site) per tadpole transport. We excluded trajectories with only one final deposition site and no detailed TT.

## Movement precision

To investigate whether frogs explored the surrounding area by taking additional detours, we estimated the precision of the frogs' orientation towards the upcoming pools during TTs, and towards the territory during the HTs. To estimate navigation precision, we calculated three different parameters: (1) the straightness coefficient (SC) of the trajectory, (2) the average angular deviation between the ideal orientation angle and consecutive tracking locations, and (3) the average normal distance of tracking locations from the straight-line path. The SC is defined as the ratio between the straight-line distance to the respective goal and the actual path distance. SC ranges from 0 to 1, with 1 indicating a perfectly straight trajectory. The angular deviations were measured as the absolute angular difference between the ideal direction (angle from each tracking location to the respective goal) and the actual direction of the individual frog's movement (angle from each tracking location to the next one). For the distance of the frog's movement from the straight-line path, we calculated the perpendicular deviation of each tracking location from the straight path. While the SC takes into account the entire trajectory at once, the average angular deviations describe the movement decisions from location to location, and the distance from the straight-line path assesses the frog's position in relation to the straight line for every location. We calculated the mean angular deviation and distance from the straight-line path by averaging all values per TTs and HTs in order to have three precision measurements per TTs and HTs.

We tested for significant goal-directed orientation using absolute angles calculated with the "as.ltraj" function from the package "adehabitatLT" (*Calenge, 2015*) per TTs and HTs using Rayleigh tests with the package "circular" (*Agostinelli & Lund, 2011*).

### Influence of pool availability on movement precision

In addition, we tested whether the precision during TTs differed (for example resulting from olfactory cues from the water), when tadpole carriers approached an available deposition

site (artificial or natural pool) or an unavailable pool site (removed or desiccated). Therefore, we compared the precision of TTs (SC, average angular deviation and average distance from the straight line path per trajectory) of frogs that were heading towards available and non-available pools. We used different GLMMs with the ''SC'' (family = beta, link = logit), ''average angular deviation'' (family = gamma, link = logit) and ''average distance to the straight-line path'' (family = gamma, link = logit) as response variable. For all three models, we used ''heading towards an available vs. non-available pool'' as explanatory variable and individual frog ID as a random effect.

Finally, we investigated potential differences in precision along the tracked HTs between frogs that either did or did not encounter a site with a non-available pool (removed pools and naturally dried-out water bodies) during prior tadpole transport. For the analysis, we used different linear mixed models (LMM), with ''average angular deviation'' and ''average distance from the straight-line path'' as response variable and a GLMM (family = beta, link = logit) with ''SC'' as response variable. For all three models, we used ''removed pool encountered during prior tadpole transport'' (yes/no) as explanatory variable and the individual frog ID as random factor to account for repeated trajectories of the same individuals.

## Model selection

All the full models but one contained only a single explanatory variable, which was a binary factor. We compared these models with the corresponding null (intercept) model based on the second-order form of Akaike's information criterion (AICc; (*Hurvich & Tsai, 1989*)). No difference between the full model and the null model (i.e., $\Delta$AICc $\leq 2$) indicates that the variation of the response variable is not explained better by the full model than by the null model (*Burnham & Anderson, 2002*). If this is the case the null hypothesis is supported and we therefore do not present the model parameters in the results. For the model with several explanatory variables we created all possible candidate models (all-subset modeling) following the information-theoretic approach (*Burnham & Anderson, 2002*). We ranked them according to their AICc values and selected those within $\Delta$AICc $\leq 2$ with respect to the top-ranked model. We estimated parameters for each explanatory variable included in the $\Delta$AICc $\leq 2$ subset by model averaging (following *Burnham & Anderson, 2002*).

All models were calculated using the R packages ''lme4'' (*Bates & Maechler, 2010*) or ''glmmADMB'' (*Bolker et al., 2014*), and the package ''MuMIn'' (*Bartoń, 2013*) for model averaging.

## RESULTS

### Frog sampling and movement analysis

During the study period, we captured 67 individual males a total of 658 times. 408 captures were associated with male territorial behavior and were used to calculate territory centers (Fig. 1). We observed 50 tadpole transports by 30 males and tagged 20 individuals, which allowed us to track 28 tadpole transports. From all tagged frogs, we used the TTs and/or HTs of 16 individuals for movement analysis: $n = 15$ TTs from 10 individuals (including five individuals with 2 TTs each); $n = 22$ HTs from 16 individuals, (including one individual

with 3 HTs and four individuals with 2 HTs each). Other trajectories were excluded because some individuals had either not moved further than 5 m for two days ($n = 1$), had an injury ($n = 1$), were predated on by a spider ($n = 1$), or the entire tadpole transport took place inside their territory ($n = 1$). All averaged values (i.e., distance, time and speed) were estimated by first averaging per trajectory section (TT and HT), followed by calculating the overall average from all TTs and HTs.

Summing up the entire trajectory (interpolated start + TT + HT; $n = 14$, 10 individuals) frogs moved an average of 141.73 m (sd = 68.87 m, range = 59.94–276.01 m) and were tracked for an average of 17.62 h (sd = 14.01, range = 6.42–58.4 h, nights excluded). Tracked TT ($n = 15$, 10 individuals, see Fig. 2) covered a distance of 56.34 m (sd = 38.09 m, range = 4.93–141.37 m) on average. The interpolated path was 39.21 m (sd = 24.72 m, range = 0–95.16 m, for $n = 12$, 7 individuals) on average, adding up to an average distance of 87.71 m (sd = 40.29 m, range = 35.32–166.43 m) for the whole TT. The elapsed time until all tadpoles were deposited was 5.55 h (sd = 2.7 h, range = 2.08–13.25 h, nights were excluded for $n = 1$) on average. The speed during tracked TTs was on average 10.16 m/h, reaching a maximum of 17.91 m/h (averaged over the entire TTs, distance = 55.16 m, time = 3.08 h; for further details see Table S3). During HTs, male frogs moved an average distance of 54.57 m (sd = 29.63 m, range = 15.98–123.46 m, $n = 22$, 16 individuals, see Fig. S2) and the average time elapsed until their return to the territory was 10.78 h (sd = 10.38 h, range = 1–49.9 h, nights were excluded for $n = 10$). Speed during homing was 7.22 m/h on average with a maximum of 22.16 m/h (averaged over the entire HTs, distance = 22.16 m, time = 1 h; for further details see: Table S4). The movements during the TTs and the HTs were characterized by stop-and-go phases varying in speed (range of speed from one tracking location to the next one = 0–70.12 m/h, for further details see Figs. S3 and S4).

Frogs moved significantly faster when tadpoles were still present compared to the subsequent homing (GLMM estimates ± standard errors: with tadpoles 0.132 ± 0.015; without tadpoles 0.092 ± 0.017; $p = 0.02$; see also Fig. 3). Results from our model investigating effects of weather on the movement speed of frogs can be found in the Tables S1 and S2).

## Pool visits

We recorded 49 pool visits during 28 TTs (20 tagged individuals) including all artificial and natural, available and removed/dry pools. Frogs carried on average 8.5 tadpoles (sd = 4.9, $n = 27$; 1 excluded) and were never observed to visit further pools after depositing all their tadpoles. In 25 of the 28 TTs, successful deposition of tadpoles was recorded. The remaining three individuals either lost the tadpoles overnight ($n = 1$), were predated on by a spider ($n = 1$) or disappeared during tracking ($n = 1$). From the 49 pool visits recorded, deposition occurred in 29 cases (artificial pools = 26 times, natural deposition sites = 3 times) and no deposition took place in 20 cases (removed artificial pool = 11 times, dry natural pool = 6 times, available artificial pool = 3 times). On average, the first observed pool visited by each frog ($n = 28$) was 41.07 m (range = 2.1–98.33 m) away from the territory center. During the tracked TTs, male frogs visited an average of 2.4 depositions

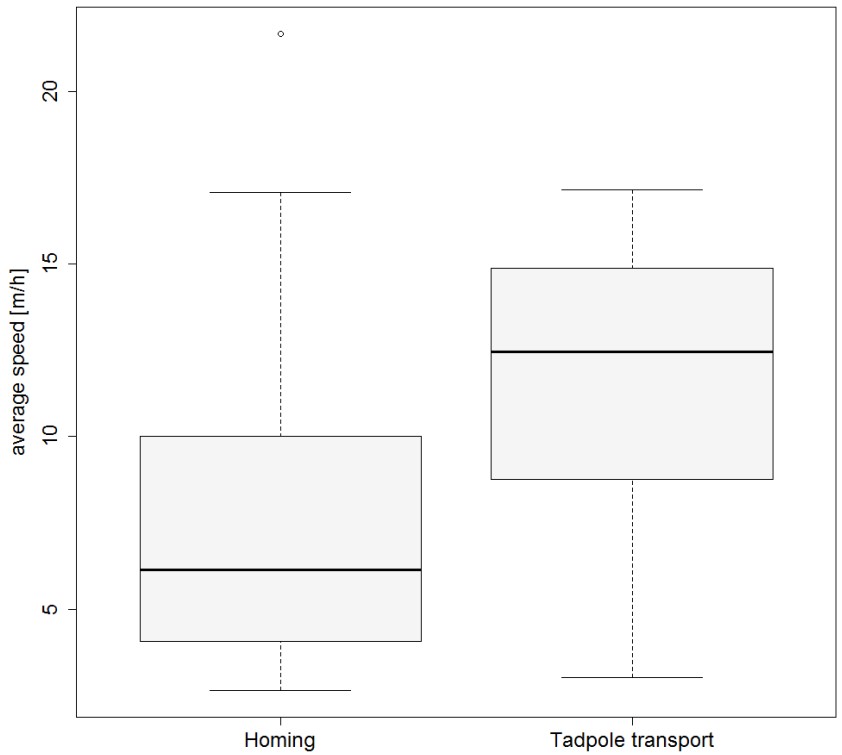

**Figure 3  Movement speed during tadpole transport and homing.** Boxplot showing the average speed (m/h) for the TTs and HTs.

sites per transport event ($n = 15$, range $= 1$–$4$, including removed and desiccated pools). However, we cannot exclude that frogs visited other deposition sites before we encountered them.

## Movement precision

Frog movement was strongly directed towards potential deposition sites and the home territory for TTs and HTs, respectively (see Figs. 2 and 4 and Fig. S2). We only used tracks with more than three locations for the analysis of directionality. The tadpole carriers ($n = 14$, 10 individuals; 1 excluded) moved directly to and between potential deposition sites with an average straightness coefficient of 0.83 (sd $= 0.13$), an average angular deviation of 24.52° (sd $= 36.58$°; Rayleigh test $p < 0.001$) and an average linear deviation of 2.9 m (sd $= 4.02$) from the straight-line path. On their way back to the home territory, frogs ($n = 22$, 16 individuals) reached an average straightness coefficient of 0.87 (sd $= 0.12$) and moved with an average angular deviation of 32.28° (sd $= 42.58$°; Rayleigh $p < 0.001$) from the ideal path and an average linear deviation of 2.04 m (sd $= 2.03$).

Movement precision during TTs: None of the models with SC, average angular deviation or average distance to the straight-line path as response variable and heading towards an available or unavailable pool as explanatory variable improved the AICc compared to their corresponding null model. Thus, we could not find any difference in the frog's movement behavior when encountering a removed or available pool (Table S5).
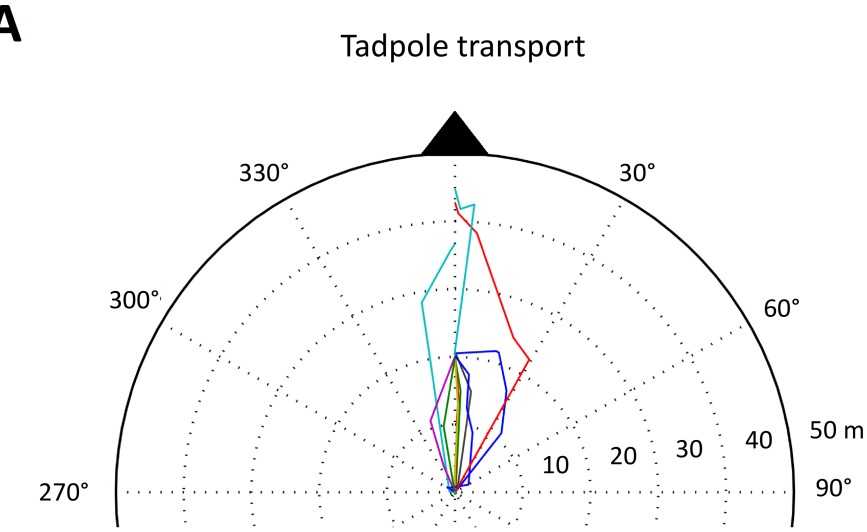

**A**

Tadpole transport

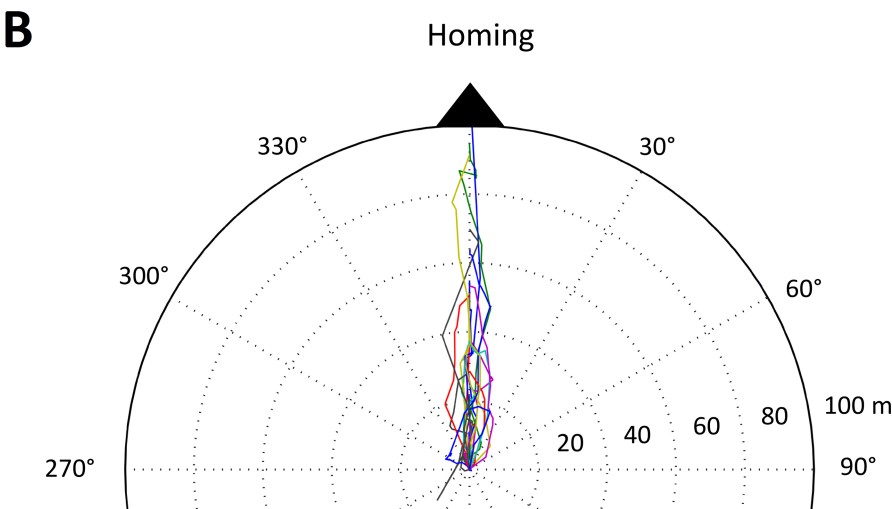

**B**

Homing

**Figure 4** **Tadpole transport and homing trajectories.** Part of a polar plot showing (A) the tadpole transport (TT) and (B) homing (HT) trajectories of male *A.femoralis*. Each colored line represents one trajectory (TT: $n = 11$, 7 individuals; HT: $n = 22$, 16 individuals). For better visualization, we only plotted the parts of TTs showing the movement between the first two pool sites visited. TTs that did not pass by at least two pool sites are excluded from the plot. All trajectories were normalized to a single starting point (center of the plot), which corresponds to the first pool visited for the TTs and to the last deposition site for the HTs. The full extent of the plot corresponds to 50 m for TTs and to 100 m for HTs.

Movement precision during homing: The two models with the response variables SC or average distance to straight line and heading towards an available or unavailable pool as explanatory variable did not improve the AICc compared to their corresponding null model. For the model with average angular deviation as response variable, the AICc of the full model was lower compared to the corresponding null model ($\Delta$AICc $= 2.59$), but there was no difference between the levels of the explanatory variable (GLMM estimates

± standard errors: heading towards an available pool 28.733 ± 4.860; heading towards a non-available pool 29.580 ± 6.625; $p = 0.9$). Overall, this indicates that there is no difference during homing between frogs that did or did not encounter an unavailable deposition site during tadpole transport (Table S6).

## DISCUSSION

### Movement patterns

In our study, we quantified movements of tadpole-transporting frogs, but we did not observe any exploratory behavior of *A. femoralis* during tadpole transport or subsequent homing. Males exhibited highly directed movement towards deposition sites and, in contrast to our predictions, also on their way back to their home territory. Frogs moved faster when transporting the tadpoles than when homing and we found no effect of pool presence (available or unavailable deposition site) on the precision or speed of movement.

For breeding males, long distance movements can have many potential costs such as energy expenditure, exposure to predation, lost mating opportunities, and the risk of losing the territory altogether (*Bell, 2012*). Straight movement towards previously learned deposition sites and back to the home territory minimizes both distance and time, thereby reducing such costs. During the TT these costs may even be higher, as offspring survival also has to be taken into account (*Downie et al., 2005*), which might explain the increased speed during tadpole shuttling. During the breeding season, particularly in the afternoons, male *A. femoralis* are mostly found in their territories, calling to attract females and repel competing males (*Kaefer et al., 2012*). Male mating success in *A. femoralis* is determined by the possession of a territory (*Ursprung et al., 2011*) and probably also by calling activity and territory size (*Roithmair, 1992*). Exploration during the tadpole transport could decrease mating success and hence the individual fitness of males. We suggest that the costs of exploratory behavior during tadpole transport outweigh potential benefits for breeding males, and thus exploration is more likely to occur when males are not currently defending a territory.

During the breeding season, new artificial pools are used for tadpole deposition within days or weeks, indicating that at least some exploration occurs during the reproductive season (*Ringler, Hödl & Ringler, 2015*; M Ringler & A Pašukonis, pers. obs., 2009–2017). Territorial displacements, as well as spontaneous territorial shifts, have been observed both within and between reproductive seasons (*Ringler, Ursprung & Hödl, 2009*), and are particularly common at the onset of reproduction (M Ringler & A Pašukonis, pers. obs., 2009–2017). Such shifts may provide opportunities to explore the surrounding area and update the information on pool availability that are less costly. In addition, very little is known about *A. femoralis* movements outside the breeding season. We have regularly observed juveniles as well as adult frogs in the immediate vicinity of water-filled artificial pools during dry periods, when calling and reproductive activity is low (M Ringler & A Pašukonis, pers. obs., 2009–2017) Sensitive learning phases during the juvenile stage are common in vertebrates (*Immelmann, 1975*), but since adult frogs can establish new territories (*Ringler, Ursprung & Hödl, 2009*) and discover new pools during

the breeding season (*Ringler, Hödl & Ringler, 2015*), the spatial learning mechanism seems to be flexible and open-ended. In addition, both complete pool desiccation during the dry season and disappearance of some pools over time suggest that exploration during juvenile dispersal and the non-reproductive season alone are unlikely to provide sufficient information to the frogs for efficient tadpole transport. Studying juvenile dispersal and the adult movement patterns outside the breeding season, however, will be necessary to fully understand when and how the frogs acquire new spatial information. Furthermore, investigating the learning mechanisms underlying the spatio-cognitive capacity could provide insights into its potential constraints and how animal movement is shaped (*Fagan et al., 2013*).

We found that movement speed was significantly higher when tadpoles were still being carried compared to the speed after tadpole deposition had occurred. As has been shown in another poison frog species (*Smith et al., 2006*), the presence of tadpoles does not seem to inhibit the locomotory performance of transporting frogs. We assume that overall costs during tadpole transport are higher than during homing, as the survival of the transported offspring also has to be accounted for. As a result, frogs appear to adjust their movements and, for instance, quickly deposit their offspring to prevent the tadpoles from drying out (*Downie et al., 2005*). However, why frogs are slower during homing still remains unclear. Since frogs do not appear to explore during homing, faster homing should reduce the risk of losing mating opportunities (*Roithmair, 1992*; *Ringler et al., 2013*), or even the entire territory. Potential exhaustion after tadpole deposition, time needed for homewards orientation or high risk of predation during fast movement might explain slower movement during homing, but future studies need to examine these factors in more detail. All movements were in general characterized by stop-and-go phases of varying duration and speed. Intermittent movement patterns can be found in many organisms ranging from protozoans to mammals, and in a variety of behavioral contexts such as searching or habitat assessment. Frequent stops could lead to perceptual benefits because animals then have time to scan the area, and conspicuousness towards predators might be reduced (*Kramer & McLaughlin, 2001*). Hence, the stop-and-go locomotion pattern of male frogs during tadpole shuttling and homing could be a further adaptation for orientation and resting, while reducing risks related to continuous movement.

## Pool visits

During the entire study, we never observed male frogs that encountered a removed artificial or dry natural pool exhibiting any exploratory behavior after the deposition of tadpoles. Two individuals that encountered an unavailable deposition site were observed during a second tadpole transport event when they visited the very same unavailable pool. In contrast to our predictions, the actual availability of potential deposition sites had little influence on the movement patterns of male frogs, and no apparent updating of information concerning resource availability during tadpole transport occurred. In our study, all frogs except one (which lost tadpoles overnight) managed to find an available pool for deposition even if they had previously encountered a removed or dry pool. Furthermore, we recorded one male visiting a natural pool site that never held water during the entire study period and could
only have been known as a potential deposition site from previous years. These findings suggest that male frogs rely predominantly on their spatial memory to find deposition sites and do not invest time and energy in exploring further pools during tadpole transport. The availability of suitable breeding pools can change rapidly in the tropics as a result of sudden, heavy rainfalls or fast desiccation due to strong solar radiation. Nevertheless, resource availability seems to be sufficiently stable for frogs to rely on memory-based orientation strategies. Previously unavailable pools might turn into available breeding resources after one heavy rainfall, whereas some pool locations remain the same not only for the entire breeding season, but for several years at a time. Thus, frogs may remember previously visited pools as having been available in the past and to be at least in principal able to contain water. The best strategy might be to remember previous sites and visit them repeatedly, even at the risk that they might have vanished since the last visit. Many nectar-feeding species use similar strategies and primarily use spatial information to relocate flowers over object-based cues (e.g., *Hurly & Healy, 2002*; *Thiele & Winter, 2005*; *Carter, Ratcliffe & Galef, 2010*). In a study on nectar-feeding bats (*Glossophaga commissarisi)* they could show that flight approaches to feeders that were primarily guided by spatial memory were of shorter duration than approaches that included object-based cues, indicating that there is a short-term energy advantage to the spatial-memory strategy (*Thiele & Winter, 2005*). Relying primarily on spatial memory to find pools repeatedly is probably also more efficient than locating them based on goal-associated cues.

We did not find any difference when comparing the precision and speed of frogs that moved towards an available or unavailable deposition site, suggesting that pool-associated cues, such as odor, do not play a major role in orientation towards the pools. This further supports evidence that *A. femoralis* mainly uses suitable pools based on spatial memory (*Pašukonis et al., 2016*). Olfaction, however, has been shown to play a role in poison frog pool choice (*R. variabilis; Schulte et al., 2011* and *Schulte & Lötters, 2014*) and orientation (*A. femoralis; Pašukonis et al., 2016*). In *A. femoralis* olfactory cues might especially play a role in the initial discovery and evaluation of suitable deposition sites. Finally, other indirect cues, such as pool-associated microhabitat (e.g., as in salamanders (*Jenkins, McGarigal & Timm, 2006*)) or calls of heterospecifics (e.g., as in newts (*Diego-Rasilla & Luengo, 2007*)), might be used by tadpole-transporting frogs to discover breeding sites. For example, transporting males of the poison frog *Dendrobates tinctorius* were found to gather at sites of fresh treefalls, which often provide new deposition sites (*Rojas, 2015*).

Recent tracking and genetic studies have revealed that *A. femoralis* remember the location of up to six different pool sites (*Pašukonis et al., 2016*) and that frogs actively partition their offspring across several water bodies as a possible reproductive bet-hedging strategy (*Erich et al., 2015*). Our results corroborate these findings as tracked individuals usually visited and used two to three deposition sites per tadpole transport event. Most frogs moved considerable distances beyond the boundaries of their territory and often used pools that were not the closest ones to their territory. However, we cannot entirely rule out that some of the movements we observed between multiple pool sites were a byproduct of disturbance from tracking and handling. If tadpole distribution over several sites improves offspring survival, then knowing more pool locations should have direct fitness consequences. We

speculate that this creates a trade-off between minimizing the costs of tadpole transport and maximizing the potential benefits gained through offspring partitioning over multiple learned sites (see *Erich et al., 2015*).

## CONCLUSIONS

While it remains unknown when and how the poison frog *A. femoralis* collects information about the surrounding area, we provide, for the first time, detailed information about the movement patterns during tadpole transport. We observed highly directional movement between territories and pools as well as between pools, suggesting an advantage of quick tadpole transport and homing over additional detours to explore the area. Future research should investigate in more detail the costs and benefits of tadpole transport in order to understand the trade-offs shaping movement strategies in such dynamic environments. Further, the mechanisms that allow poison frogs to establish a spatial memory and orientate with such high precision in the rainforest remain unknown.

Despite extensive capture-recapture studies (e.g., *Brown, Morales & Summers, 2009*; *Ringler, Ursprung & Hödl, 2009*), and some tracking of *A. femoralis* after translocations (*Pašukonis et al., 2014a*; *Pašukonis et al., 2014b*) and tadpole transport (*Pašukonis et al., 2016*), still very little is known about the natural movement patterns of poison frogs and tropical amphibians in general. Tropical amphibians exhibit a huge diversity in breeding strategies ranging from explosive breeders that gather in ponds to prolonged breeders that depend on widespread, ephemeral pools for tadpole development. This diversity and the dependence on water bodies for reproduction make tropical amphibians a valuable study system to investigate how animals deal with varying resource availability and how this shapes movement patterns. Our findings contribute to the knowledge of spatial behavior in poison frogs and will hopefully encourage further research on movement ecology of tropical amphibians.

## ACKNOWLEDGEMENTS

We are grateful to the staff of CNRS Guyane and the Nouragues Ecological Research Station for logistic support in the field and to Nicolas Perrin for providing the RECCO® transceiver. We also thank Rosanna Mangione and Steffen Weinlein for their help in the field, Gesche Westphal-Fitch for proofreading and correcting language and style of the manuscript, Lisa Maria Schulte, Rick Lehtinen, and one anonymous reviewer for constructive feedback on the manuscript.

### Funding

This study was funded by the Austrian Science Fund (FWF) projects W1234-G17, P24788-B22 (PI: Eva Ringler), the University of Vienna (KWA grant and Förderungsstipendium to KB), and the Ethologische Gesellschaft e.V. master's thesis grant (KB). AP and the meteorological data were funded by "Investissement d'Avenir"

grants (including Nouragues Travel Grant 2015) managed by Agence Nationale de la Recherche (ANAEE-France: ANR-11-INBS-0001; CEBA: ANR-10-LABX-25_01). AP (FWF: J3827-B29) and MR (FWF: J3868-B29) were also supported by Erwin Schrödinger fellowships from the Austrian Science Fund. There was no additional external funding received for this study. The funders had no role in study design, data collection and analysis, decision to publish, or preparation of the manuscript.

## Grant Disclosures

The following grant information was disclosed by the authors:
Austrian Science Fund: FWF: W1234-G17, P24788-B22.
Erwin Schrödinger fellowships: FWF: J3827-B29, FWF: J3868-B29.
University of Vienna (KWA grant and Förderungsstipendium).
Ethologische Gesellschaft e.V. (master's thesis grant).
Investissement d'Avenir, Agence Nationale de la Recherche: ANAEE-France: ANR-11-INBS-0001, CEBA: ANR-10-LABX-25_01.

## Competing Interests

The authors declare there are no competing interests.

## Author Contributions

- Kristina B. Beck conceived and designed the experiments, performed the experiments, analyzed the data, wrote the paper, prepared figures and/or tables, reviewed drafts of the paper.
- Matthias-Claudio Loretto analyzed the data, wrote the paper, reviewed drafts of the paper.
- Max Ringler analyzed the data, contributed reagents/materials/analysis tools, wrote the paper, reviewed drafts of the paper.
- Walter Hödl conceived and designed the experiments, contributed reagents/materials/analysis tools, reviewed drafts of the paper.
- Andrius Pašukonis conceived and designed the experiments, analyzed the data, contributed reagents/materials/analysis tools, wrote the paper, prepared figures and/or tables, reviewed drafts of the paper.

## Animal Ethics

The following information was supplied relating to ethical approvals (i.e., approving body and any reference numbers):

Our study was approved by the scientific committee of the Nouragues Ecological Research Station. All necessary permissions were provided by the Centre National de la Recherche Scientifique (CNRS) and by the Direction Régionale de l'Environnement, de l'Aménagement et du Logement. No IRB approval was necessary as per the current regulations of the University of Vienna at the time when fieldwork was conducted.

## Data Availability

The raw data has been uploaded as a Supplemental File.

## Supplemental Information

Supplemental information for this article can be found online at http://dx.doi.org/10.7717/peerj.3745#supplemental-information.

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
