# Peer review of "Relying on known or exploring for new? Movement patterns and reproductive resource use in a tadpole-transporting frog"

_PeerJ, doi:10.7717/peerj.3745_

## Round 0.1 · original submission · Minor Revisions

Hi, please revise your manuscript according to the reviewers' suggestions. Thanks for sending an excellent manuscript!

·

Basic reporting

This is a very nice study giving some new information about movement patterns in poison frogs during and after tadpole transport. The manuscript has a very clear structure, is well written and develops around a well-thought-out hypothesis. The introduction gives a good background regarding animal movement decisions in general as well as poison frog behaviors in particular. However, the discussion could be embedded in a slightly broader context (see general comments section). The figures and tables are well made and help to understand the article. I only suggest one potential extra figure for the supplementary material (see general comments section). The references given are well chosen, but I do have two small suggestions where further literature could be cited (see general comments section). Raw data are shared.

Experimental design

The experiments conducted in this study are well thought through and the methods are described in sufficient detail. The methods follow a high standard, both during fieldwork as well as the statistical analysis (I only have two minor questions regarding the methods, see general comments section). The research conducted in this study should be relevant for many readers that are interested for example in animal movements, parental care or homing behaviors, and falls in my opinion into the aims and scope of the journal.

Validity of the findings

In my opinion the findings of this study are very interesting, the data are robust and statistically sound and the conclusions are well drawn.

Additional comments

Introduction
-Line 81-82: „ While complex parental behavior in this group of frogs has attracted a considerable amount of research (for review, Wells, 2007)…” > considering that there has been a lot of research regarding parental care behavior in poison frogs in the last 10 years, the citation of a newer review or a couple of examples would be more appropriate here
-Line 90-91: “Recent studies have shown …. Together these results suggest that poison frogs rely on spatial memory to successfully navigate in their environment…” > You might also want to add some general examples of poison frogs maintaining home ranges, since this behavior also already gives a hint that they have spatial memories (e.g. Brown et al. 2009 in Animal Behaviour, Werner et al. 2011 in Journal of Natural History or Neu et al. 2016 in South American Journal of Herpetology)
-Line 99: “Males use multiple, scattered aquatic sites outside their territories…” Please clarify: do males only deposit tadpoles outside their territories? And does that mean that they deposit them in other males territories or do they specifically search for unoccupied areas? Can you give a rough estimate about the size of a typical male´s territory?

Methods
-Line 178ff: Since it was not mentioned yet at this point how big the territories are, please explain why it was easier/more efficient to find and start tracking the tadpole-carrying frogs at a pool instead of starting from their territories. You might also want to mention that the clutches are not easy to find, and that for this reason you did not start tracking at the deposition sites.
-Line 261-264: “To assess whether frogs usually choose the closest pool to their territory, we compared the distances from the respective territory centers to the first pool a male visited to the distance to the nearest pool site (all natural and artificial deposition sites were taken into account, regardless of whether sites were available or not)” > How did you know that the frogs did not visit pools before you found them (especially if pools were not available, i.e. if you could not define a non-visited pool by the lack of tadpoles)? Also compare your statement in line 368: “However, we cannot exclude that frogs had visited other deposition sites before being encountered.”

Results/Figures:
-Line 373-373: “Frog movement was strongly directed towards potential deposition sites and the home territory for TTs and HTs, respectively (see Fig. 2, 3 and Fig. S2)” > Would it be possible to show the directional movement between pools as a polar plot as well (maybe in the supplementary material and for example just between the first two pools each frog visited)? This way the figures for HT (i.e. Fig. 3) and TT would be better comparable.
-Fig. 1: are the asterisks not rather the territory centers than the whole territories?

Discussion
-The discussion is generally very nice and covers all potential scenarios/questions regarding tadpole transport and homing behavior in A. femoralis. However, the section where the results of this study are put in a wider context is extremely short (and only covers the stop-and-go phases of the frogs but not the actual study results). A couple of sentences why this study is also important for non-poison-frog-researchers would complete this manuscript.

·

Basic reporting

Everything is OK here, no problems.

Experimental design

Everything is OK here, no problems.

Validity of the findings

Everything is OK here, no problems.

Additional comments

Tadpole transport is a relatively rare form of parental care in frogs and, as the authors state, we really don’t know very much in detail about what actually happens before, during and after tadpole transport events. Using a combination of innovative technology and old fashioned field work, this study starts to answer some of these long-standing questions. While I am not an expert in all of the statistical approaches used, the analyses seemed appropriate and carefully done. All graphs and tables are clear and useful to describe the results. The literature cited is relevant and includes both recent and classic papers. I found this manuscript overall to be a very interesting and useful contribution to the literature.

A few detailed comments follow:
The statistical methods section is a bit dense. Perhaps subheadings in this section would make the purpose of each analysis a bit more clear?
Line 329: the text “with five times 2 trajectories per individual” is a bit unclear to me.
Line 330 “with once 3 trajectories per individual and four times 2 trajectories per individual” – same comment
Line 393 missing end parenthesis
Line 441 adaption = adaptation?

Reviewer 3 ·

Basic reporting

no comment

Experimental design

no comment

Validity of the findings

no comment

Additional comments

This manuscript, which was very well written and easy to read, succeeds in setting up the goal of understanding how removal (“desiccation”) of tadpole rearing sites influences the likelihood of seeing “exploration” behavior for new such sites by father frogs carrying tadpoles. PeerJ is a good target journal for this article, because the data is important in a comparative context and will benefit the community to have it publicly available in the interest of advancing the field. These data provide strong support for the idea that frogs have a mental map of the sites where they can rear tadpoles, and in the moment that they need to deposit a tadpole they simply visit those sites until they find one that is available and usable.

In general, a few changes could be made to improve the clarity of the manuscript especially at the end of the intro and in the discussion. It is recommended the authors reduce the assumption in the text that the behaviors seen so clearly indicate that frogs are benefiting themselves or their tadpoles by moving faster (although it isn’t clear what is the relative thing that they are moving faster than). Also, the authors should discuss the possibility that frogs learn (add new information to their memory) at a time or season different than when this study was performed, and simply aren’t or cannot create new spatial memories while acting as parents (which is different from a particular cost or benefit due to not exploring at that time). Some minor suggestions to improve the manuscript are outlined below.

Lines 85-86: Consider also citing the thesis of Maple 2002 here.

Lines 86-89: Work by Schulte et al. 2011 and Stynoski et al. 2009 is also relevant to understanding the cues that poison frogs use to identify tadpole rearing sites.

Lines 93-94: Consider also citing Brust 1990’s thesis.

Line 112: This “link” is not very clear. There could just as easily be a restriction on how far fathers can put tadpoles due to limitations of spatial memory. It would help to explain better what is meant by this, or just mention that the tadpole transport distance and spatial memory limits are similar. But, suggesting this causational relationship due to a correlational piece of information is implying too much.

Line 113: This sentence is written a bit awkwardly, I had to read it a few times to understand.

Lines 125-135: This list of goals for the study is clear, and they are worthy goals to explore. But, the paragraph winds around a little bit, and could perhaps be restructured in a way that would allow for the reader to follow a more linear train of thought among those goals.

Lines 143-148: Probably worth citing here any data available on this species regarding how many tadpoles are transported at a time and if they are placed singly or in multiples (for easy comparison among other studies and species).

Line 153: The word “permit” might be appropriate instead of “permissions”.

Line 157: Remove the word “and” and replace with a comma.

Line 158: Not clear how the cross-shaped rearrangement is relevant to the study? To make the arrangement more obvious to the reader before they reach the end of the paragraph, suggested to reference Figure 1 earlier in this paragraph (perhaps around line 158?).

Line 207: This sentence has problems with syntax, editing suggested.

Lines 328-331: This information breaking down how many observations of what type were made per group is very confusing to read.

Line 399: Similarly to the comment for the last paragraph of the introduction, a summary sentence or two to start the discussion that can wrap up the overall objective and take home message of the study would be beneficial before diving into each section of the details of the findings.

Line 410: Another possibility is that male trajectories to places where they can deposit offspring are based on memories of natal pools and dispersal. This option and any data available to support or reject that hypothesis would benefit this discussion.

Line 430-433: This statement seems like a large assumption to make without data to support it. In fact, exactly what you suggest would be a great way for frogs to learn about the likely places they will be able to place tadpoles. Seasonal dryness or what is meant by “natural dynamics of small natural pools” don’t offer obvious reasons that frogs could not gain lots of information this way.

Line 435-442: This paragraph seems like an extensive speculation. What data from this study have specifically augmented this point of view with new information that make it worth describing a whole paragraph about this idea?

Line 470-471: You suggest that the weather changes “rapidly” in the tropics. Rapid relative to what? It isn’t clear that it is in fact more rapidly changing or unpredictable than elsewhere globally. This point is not convincing.

Line 477: These results should be compared/contrasted with the findings from Schulte et al. 2011, Stynoski et al. 2009, and perhaps also Rojas 2016.

Lines 482-484: Its possible that adding this information sooner in the paper would be beneficial for the reader to interpret the results as they are reading about them. It is important context for understanding the ways that natural selection and spatial memory could be functioning to allow males to make decisions about tadpole depositions.

Line 501: “potential influencing factors” as written isn’t clear what it is referring to, and is vague.

Line 502: You suggest that there is a cost-benefit reason that the frogs deposit tadpoles and return to their territories relatively quickly. While this could be true, there are many other possible explanations for the behavioral pattern that you noted. For example, the mechanism for spatial memory could be such that frogs can only learn or memorize sites during certain seasons or developmental phases, which could be a simple product of neurobiological limitations rather than an optimized cost-benefit for the behavior. Changing the way this is written would allow for these other possibilities if future fitness studies were to show that there isn’t actually a clear cost or benefit to the speed of transport, but its simply what frogs are or aren’t capable of.

Line 509-510: This statement about modeling doesn’t appear to have much of a connection to your work and also isn’t a very strong way to end the manuscript. Suggested to remove and end at the preceding sentence.

Table 1: This data would likely be more informative as a box plot rather than a table.

Figure 1: Would be informative to include the average territory size in the caption for this figure.

---

## Round 0.2 · accepted · Accept

Thank you for your careful revision of the manuscript, and congratulations on a fine paper!